# Exercise-Intervened Endothelial Progenitor Cell Exosomes Protect N2a Cells by Improving Mitochondrial Function

**DOI:** 10.3390/ijms25021148

**Published:** 2024-01-17

**Authors:** Shuzhen Chen, Smara Sigdel, Harshal Sawant, Ji Bihl, Jinju Wang

**Affiliations:** Department of Biomedical Sciences, Joan C. Edwards School of Medicine, Marshall University, Huntington, WV 25755, USA; chens@marshall.edu (S.C.); sigdels@marshall.edu (S.S.); sawantha@marshall.edu (H.S.); bihlj@marshall.edu (J.B.)

**Keywords:** exosomes, EPC-EXs, exercise, hypertension, hypoxia, miR-27a

## Abstract

We have recently demonstrated that exosomal communication between endothelial progenitor cells (EPCs) and brain endothelial cells is compromised in hypertensive conditions, which might contribute to the poor outcomes of stroke subjects with hypertension. The present study investigated whether exercise intervention can regulate EPC–exosome (EPC-EX) functions in hypertensive conditions. Bone marrow EPCs from sedentary and exercised hypertensive transgenic mice were used for generating EPC-EXs, denoted as R-EPC-EXs and R-EPC-EX^ET^. The exosomal microRNA profile was analyzed, and EX functions were determined in a co-culture system with N2a cells challenged by angiotensin II (Ang II) plus hypoxia. EX-uptake efficiency, cellular survival ability, reactive oxygen species (ROS) production, mitochondrial membrane potential, and the expressions of cytochrome c and superoxide-generating enzyme (Nox4) were assessed. We found that (1) exercise intervention improves the uptake efficiency of EPC-EXs by N2a cells. (2) exercise intervention restores miR-27a levels in R-EPC-EXs. (3) R-EPC-EX^ET^ improved the survival ability and reduced ROS overproduction in N2a cells challenged with Ang II and hypoxia. (4) R-EPC-EX^ET^ improved the mitochondrial membrane potential and decreased cytochrome c and Nox4 levels in Ang II plus hypoxia-injured N2a cells. All these effects were significantly reduced by miR-27a inhibitor. Together, these data have demonstrated that exercise-intervened EPC-EXs improved the mitochondrial function of N2a cells in hypertensive conditions, which might be ascribed to their carried miR-27a.

## 1. Introduction

Neurological disorders are one of the major causes of death and morbidity, with stroke being the fifth leading cause of death in the United States. In the clinic, protecting brain cells, such as neurons, from ischemic injury and promoting neurological recovery are critical in stroke management. Hypertension and physical inactivity are common cerebrovascular risk factors. Regular physical activity could delay the development of hypertension [1] and decrease the incidence of cerebrovascular diseases, including stroke and vascular dementia [2,3]. Studies have found that physical exercise can elicit a more pronounced effect on lowering blood pressure in hypertensive patients than in normotensive individuals [4]. Other reports showed that exercise intervention could inhibit the progression from normal conditions to pre-hypertension and from pre-hypertension to hypertension [5,6], suggesting that exercise could be a therapeutic tool to manage hypertension, especially in high-risk populations such as individuals with a family history of hypertension. However, despite the inspiring evidence derived from epidemiological and interventional trials, the underlying molecular mechanism of exercise intervention on hypertension-related cerebrovascular diseases such as ischemic stroke is not fully understood.

Endothelial progenitor cells (EPCs) are circulating precursors of endothelial cells derived from bone marrow. EPCs play a vital role in protecting against endothelial dysfunction and cardiovascular diseases [7]. The number and function of circulating EPCs are altered in many cardiovascular diseases, although the pathophysiological significance of these variations is not yet entirely clear. Luo and colleagues showed that the regenerative and reparative capability of EPCs declined in hypertensive conditions [8]. Marketou et al. reported that EPCs are involved in arterial stiffness and remodeling in hypertensive patients [9], suggesting EPCs might be used as a therapeutic target. Indeed, Medeiros and colleagues found that a phenolic monoterpene called carvacrol can improve vascular function in hypertensive rats by improving the EPC migration, increasing colony-forming units, and reducing reactive oxygen species production [10]. Numerous studies have also demonstrated that physical exercise can stimulate the mobilization of EPCs and promote EPC differentiation, thereby enhancing the angiogenesis capability of EPCs and contributing to vascular repair [11]. A recent clinical study reveals that continuous exercise training consisting of high-to-moderate intensity, adequate duration, and combined training with aerobic and resistance exercise stimulates EPC mobilization in patients with cardiovascular diseases in addition to healthy individuals [12]. Together these findings suggest that EPCs do react to exercise intervention.

Exosomes (EXs) are major extracellular vesicles measuring ~10–150 nm in diameter. During the past two decades, there has been an accelerated interest in EX research due to their putative roles in physiology, pathophysiology, diagnostics, drug delivery, and possible applications as new therapeutic compounds [13,14,15]. Increasing evidence suggests that the cargoes and functions of EXs vary upon their cellular origin and stimulation status. EXs released under exercise intervention conditions have been suggested as novel players that mediate cell–cell communication by promoting the systemic beneficial effects of physical activity [16]. Fruhbeis and colleagues found EX release increases in response to exercise training in an acute intensity-dependent manner in humans [17]. Another study using the diabetic db/db murine model of type 2 diabetes found cardiac-derived EXs containing several miRNAs (miR-455, miR-29b, miR-323-5p, and miR-466) were released in response to acute endurance exercise [18]. Additionally, exercise intervention has also been shown to regulate the function of EXs derived from EPC (EPC-EXs) in normal C57BL/6 mice [19,20]. More recently, a preclinical report shows that exercise-derived extracellular vesicles exhibit anti-anxiety effects [21]. All these findings suggest that EXs respond to the exercise intervention. Our recent study showed that the communication between EPC-EXs and brain microvascular endothelial cells was compromised in hypertensive and hypoxia conditions [22]. However, whether exercise intervention could affect the functions of EPC-EXs in hypertensive conditions is largely unknown.

As we know, hypertension development is induced by the disturbance of the renin–angiotensin–aldosterone system [23]. Angiotensin II (Ang II), an octapeptide produced from the substrate angiotensinogen, is a key factor in the renin–angiotensin–aldosterone system. Early studies have suggested that overexpression of Ang II contributes to the progress of ischemic stroke in mice [24]. In the present study, we assumed that exercise intervention can influence the exosomal communication of EPC-EXs with neurons in an in vitro cell culture system challenged by Ang II and hypoxic conditions. We applied Ang II to neurons to mimic the hypertensive conditions. Meanwhile, stroke-like conditions were achieved by the oxygen–glucose deprivation method to create an ischemic environment in the cell culture system [25]. Using the constructed hypertensive plus ischemic-like cell injury models, we tested the functions of EPC-EXs harvested from hypertensive transgenic mice that were subjected to either physically active or sedentary lifestyles. Meanwhile, we analyzed the miR profiles and studied their potential roles of EPC-EXs released under hypertensive conditions.

## 2. Results

### 2.1. Exercise Intervention Improved the Incorporation Ability of R-EPC-EXs by N2a Cells

Both R-EPC-EX and R-EPC-EX^ET^ expressed the exosomal markers (CD63, Tsg 101) and EPC-specific marker (CD34, VEGFR2) as reported (Chen et al., 2022) [22]. Before the co-culture experiment for the in vitro track, all EPC-EXs were labeled with a red fluorescent dye PKH26. As shown in Figure 1, after 8 h co-culture, a small portion of EXs was uptaken by N2a cells which was reflected by the fluorescence signal in the cytoplasm of cells treated by EXs. After 16 h co-culture, the fluorescence signals in the cytoplasm of N2a cells were remarkably increased in both groups. We also observed more R-EPC-EX^ET^ was incorporated into N2a cells—as reflected by a stronger green fluorescence signal in the cytoplasm of N2a cells—than that of R-EPC-EXs. Similar results were observed between the N2a cells treated by R-EPC-EX^ET^ and R-EPC-EXs in 24 h co-incubation. These data suggest that exercise intervention can improve the incorporation efficiency of EPC-EXs by neurons in hypertensive conditions.

### 2.2. Exercise Intervention Raised miR-27a Level in EPC-EXs of Hypertensive Mice

Accumulating evidence suggests that miRs are one of the major executors of EXs [26,27]. According to the miRNome profile data, 30 miRs were differentially expressed among 709 miRs from the EPC-EXs in exercised hypertensive transgenic mice (R+ mice) (Figure 2A). The miR-27a stood out among those miRs as the most potent neurovascular protective molecule [28,29]. Therefore, we further performed qRT-PCR analysis on miR-27a. We assessed the miR-27a level in EPC-EXs from the transgenic hypertensive R+ mice and the wild-type control mice. The data show that the miR-27a level in R-EPC-EXs was downregulated by ~2.8 fold as compared to that of the wild-type control mice. Interestingly, exercise intervention restored the miR-27a levels in R-EPC-EXs isolated from the transgenic hypertensive R+ mice (Figure 2B). This finding supports the concept that exercise intervention might alter the functions of EPC-EXs in hypertensive conditions.

### 2.3. Exercise-Intervened EPC-EXs Conveyed miR-27a and Alleviated Ang II Plus Hypoxia-Compromised Cellular Viability of N2a Cells

We conducted a co-culture experiment with N2a cells to test whether the functions of R-EPC-EX could be regulated by exercise intervention. After 24 h co-culture of R-EPC-EX or R-EPC-EX^ET^ with Ang II plus hypoxia-challenged N2a cells, the miR-27a levels in N2a cells were determined by qRT-PCR. As depicted in Figure 3A, compared to the non-treated cells, there was no significant difference in the miR-27a level of N2a cells treated by R-EPC-EX, but the R-EPC-EX^ET^ treatment increased miR-27a level by two-fold in N2a cells. These data suggest that R-EPC-EX^ET^ can convey the carried miRs to recipient cells.

To further determine whether there is a function difference between the R-EPC-EX and R-EPC-EX^ET^, we determined the cell viability of N2a cells using an MTT assay. The data showed that Ang II plus hypoxia remarkably reduced the viability of N2a cells. Both R-EPC-EX and R-EPC-EX^ET^ co-culture rescued the N2a cells as evidenced by a higher cellular viability than the vehicle (control) group. Meanwhile, R-EPC-EX^ET^ exhibited a better effect than R-EPC-EX (Figure 3B). To explore whether the boosted effect of R-EPC-EX^ET^ was ascribed to their carried miR-27a, we treated the cells with a miR-27a-specific inhibitor. The results showed that miR-27a inhibitor blocked the effects of R-EPC-EX^ET^, suggesting that exosomal miR-27a contributes to the observed effects.

### 2.4. Exercise-Intervened EPC-EXs Eased Oxidative Stress in N2a Cells Challenged by Ang II plus Hypoxia

Oxidative stress refers to raised intracellular levels of reactive oxidative species (ROS), which can damage proteins, lipids, and DNA of cells. The DHE is used as a fluorescent probe to detect ROS generation and is specific for superoxide and hydrogen peroxide. In this study, we determined the anti-oxidative stress ability of exercise-intervened EPC-EXs in N2a cells. Our data (Figure 3C,D) showed that Ang II plus hypoxia challenge significantly raised ROS overproduction by 1.6-fold. Both R-EPC-EX and R-EPC-EX^ET^ reduced ROS overproduction. Compared to the effects elicited by R-EPC-EX, R-EPC-EX^ET^ profoundly decreased ROS production as evidenced by the decreased red fluorescence signal in the nucleus of the cells.

### 2.5. Exercise-Intervened EPC-EXs Altered the Mitochondrial Membrane Potential in Neurons Challenged by Ang II Plus Hypoxia

Oxidative stress can induce mitochondrial DNA mutations and damage the mitochondrial respiratory chain. The membrane JC-1 dye is widely used to monitor mitochondrial health by detecting the mitochondrial membrane potential [30]. JC-1 dye exhibits potential-dependent accumulation in mitochondria, indicated by green fluorescence emission at (~529 nm) for the monomeric form of the probe, which shifts to red (~590 nm) with a concentration-dependent formation of red fluorescent J-aggregates. Our data (Figure 4) showed that Ang II plus hypoxia remarkably led to mitochondrial depolarization, as evidenced by a decreased fluorescence ratio of red to green. After being treated with the R-EPC-EX, the ratio of red to green was raised. An even better effect was observed in the cells treated with exercise-intervened EPC-EX.

### 2.6. Exercise-Intervened EPC-EXs Altered the Expressions of Nox4 and Cytochrome c in N2a Cells Challenged by Ang II Plus Hypoxia

Loss of mitochondrial membrane potential is a sign of bioenergetic stress and may cause cell death. We then detected the expressions of Nox4 and apoptosis protein cytochrome c in the N2a cells after the EX treatment. Our data (Figure 5) showed that Ang II plus hypoxia upregulated the expressions of Nox4 and cytochrome c in N2a cells. Compared to the effects induced by R-EPC-EX, exercise-intervened EPC-EX substantially decreased Ang II plus hypoxia-induced upregulations of cytochrome c and Nox4. Meanwhile, we observed that the effect elicited by exercise-intervened EPC-EX was inhibited by the miR-27a inhibitor.

## 3. Discussion

In this study, we first addressed that exercise intervention can regulate the functions of bone marrow-derived EPC-EXs in hypertensive conditions. The exercise-intervened EPC-EXs exhibited an improved incorporation efficiency, associated with an enhanced anti-oxidative effect on neurons via improving mitochondrial function. The underlying mechanism is majorly ascribed to their carried miR-27a.

Intercellular talk is fundamental to maintaining the homeostasis and function of the brain [31]. As novel intercellular communicators, EVs convey biological materials (e.g., miRNAs, long-noncoding RNAs, proteins) to recipient cells [32,33]. During exercise, extracellular vesicles including EXs have been suggested to provide a means for tissue crosstalk [34], which might be driven by a spectrum of bioactive molecules released into the extracellular vesicle [19,35,36]. In ischemic stroke, the communications between EPCs migrated from bone marrow and brain cells, such as endothelial cells and neurons, contribute to the recovery of neurological function [37]. In the clinic, stroke patients with hypertension have a poor outcome, though the underlying mechanism is unclear. We have previously reported that moderate exercise can regulate the release of bone marrow EPC-EXs and alleviate ischemic stroke damage in C57BL/6 mice [19,20]. Additionally, an in vitro study revealed that the exosomal communication ability of EPC-EXs with brain microvascular endothelial cells was compromised in hypertensive conditions [38]. Together these findings encouraged us to study whether exercise intervention can modulate the functions of EPC-EXs in hypertensive conditions. After the hypertensive transgenic mice (R+) mice underwent an 8-week treadmill exercise, we monitored the blood pressure of the exercised mice by the tail-cuff method. We only observed a decreased trend but no significant change in blood pressure after exercise intervention. The radiotelemetry method will be applied to monitor arterial blood pressure changes in future studies.

Among the exosomal bioactive molecules, miRs are considered one of the major executors of EXs [26,27]. Previous studies show that exercise can induce changes in the proteomic [36] or miR profiles [19,35] of small extracellular vesicles. To determine whether exercise intervention can alter the miR cargoes in the bone marrow EPC-derived EXs in hypertensive conditions, we conducted the mmu–miRome analysis. Our data reveal that the eight-week treadmill exercise did affect the packaging of miRs in EPC-EXs. We observed more than 30 miRs were either upregulated or downregulated in the bone marrow-derived EPC-EXs in the exercised hypertensive transgenic mice as compared to that in the sedentary ones. Among these miRs, the miR-27a is one of the most potent neurovascular protective miRs. Previous reports reveal that miR-27a can alleviate brain injury in the hemorrhagic stroke [29], indicating its participation in ischemic injury. MiR-27a can also alleviate hypoxia-induced apoptosis in mouse brains [38]. These findings suggest that miR-27a might be a potential therapeutic target for treating ischemic injury. Further, it is reported that miR-27a can mediate the protective effect of exercise on the aorta [28], suggesting the involvement of miR-27a in exercise intervention. Indeed, as confirmed by the qRT-PCR, the miR-27a expression was reduced in the bone marrow-derived EPC-EXs of hypertensive transgenic mice. Of note, exercise intervention remarkably restored its level, suggesting the eight-week exercise regimen promotes the package of miR-27a into EPC-EXs, although the underlying mechanism responsible for the packing requires further investigation.

Our previous study revealed that the exosomal communication between EPCs and brain endothelial cells was compromised in hypertensive conditions [22], which might contribute to the poor outcome of stroke in hypertensive individuals. Given that EX function varies on their cellular origin and cargo, and that we have observed the exosomal miR profile changed, we further determined whether the eight-week treadmill exercise intervention affected the functions of EPC-EXs released from the hypertensive transgenic mice. To achieve this, we first examined the incorporation efficiency of EPC-EXs. Using an in vitro co-culture system, we determined the uptake of EPC-EXs derived from sedentary and exercised hypertensive transgenic mice. Surprisingly, we found that the uptake efficiency of EPC-EXs by neurons from exercised mice was significantly improved during the 16–24 h co-culture period, indicating that the eight-week exercise regimen can alter the exosomal uptake. According to previous findings, several types of EX uptake mechanisms including fusion, receptor-mediated endocytosis, micropinocytosis, and phagocytosis might be involved in EX incorporation in different cellular systems [39], and individual pathway inhibitors could be applied to study the specific uptake pathway. Studies regarding the roles of the carried miRs such as miR-27a in the exosomal uptake by neurons are ongoing.

Previous preclinical studies have indicated that EXs could exhibit anti-oxidative stress ability in the neural and circulatory systems [40]. Neurons are particularly vulnerable to oxidative stress which is a process featuring excess ROS accumulation [41]. In the hypertensive and hypoxia cellular injury model, our data show that the cellular viability was compromised, and ROS production was remarkably raised, as detected by DHE staining. In this study, according to the data, we found that R-EPC-EXs did improve cell survival ability and ease oxidative stress, and the anti-oxidative effects were further enhanced by R-EPC-EX^ET^. These findings indicate that exercise intervention boosts the anti-oxidative stress capability of EPC-EXs. As discussed, exercise intervention boosted the package of miR-27a into EXs. Using the co-culture system, R-EPC-EX^ET^ conveyed their carried miR-27a to the recipient N2a cells. To check the possible roles of miR-27a, a miR-27a inhibitor was applied to block the function of miR-27a. With that, we found the effects induced by R-EPC-EX^ET^ were greatly limited, indicating that miR-27a is one of the major executors for R-EPC-EX^ET^. We will continue to investigate the roles of endogenous miR-27a in basal function and survival of the N2a cells in unstressed conditions. In the present study, we also noticed that the levels of other miRs such as miR-96 and miR-466d were significantly decreased in R-EPC-EXET. Previous studies have indicated that inhibition of miR-96 could increase the level of glutathione in neuroblast cells [42], supporting our finding. However, the exosomal miR-96 and miR-466d participation in the effects elicited by R-EPC-EXs in the N2a cells remains to be determined.

It’s well-known that mitochondria are fundamental for metabolic homeostasis in all multicellular eukaryotes including neurons. Mitochondria serve as an important hub for energy production [43]. In the event of ischemic stroke, brain mitochondria begin to lose electrochemical proton gradients, causing cessation of adenosine triphosphate synthesis and overproduction of ROS that eventually led to cell death [44]. Since we observed that R-EPC-EXs exhibited an anti-oxidative effect on N2a cells, we wondered whether this effect was related to mitochondrial function. Using the hypertensive and hypoxia cell injury model, we observed that the mitochondrial membrane potential was changed in the N2a cells as revealed by JC-1 staining [30]. Our results showed that Ang II plus hypoxia injury remarkably induced mitochondrial depolarization, as indicated by a decrease in the red/green fluorescence intensity ratio, while R-EPC-EX treatment improved Ang II plus hypoxia-induced mitochondrial membrane potential. The R-EPC-EX^ET^ treatment elicited a better effect than R-EPC-EX on modulating the mitochondrial membrane potential, which was blocked by the miR-27a inhibitor. These data are consistent with the findings of ROS production and suggest that the anti-oxidative function of R-EPC-EX^ET^ might be via regulating the mitochondrial membrane potential in this in vitro model.

Previous reports show that changes in the membrane potential could be associated with the opening of the mitochondrial permeability transition pore, allowing the passage of ions and small molecules. The resulting equilibration of ions leads to the decoupling of the respiratory chain and the release of cytochrome c into the cytosol [45]. Thus, we detected the expressions of cytochrome c and Nox4 in N2a cells. Indeed, Ang II plus hypoxia stimulus significantly raised the expressions of cytochrome c and Nox4, which is in line with ROS overproduction and decreased cell viability. R-EPC-EX rescued the cells by downregulating ROS expression, associated with a decreased cytochrome c level. The R-EPC-EX^ET^ exhibited a profound effect which was blocked by the miR-27a inhibitor. This was supported by previous reports showing that miR-27a can protect neurons by regulating the inflammatory signal pathway which is a potential target for inhibiting apoptosis and ischemic cell death [38,46,47].

## 4. Materials and Methods

### 4.1. Animals

The strain of human renin hypertensive transgenic (R+) mice (129S/SvEv-Tg; Alb1-Ren; 2Unc/CofJ) was purchased from the Jackson Laboratory (Bar Harbor, ME, USA). All animals were bred in the Animal Facilities Resources at Marshall University. The R+ mice were used as donors for bone marrow EPCs. The mice were housed in an animal facility conditioned with 12 h light/dark cycles and allowed free access to normal chow and water. They were euthanized at the age of 15–16 weeks old. Body weights were recorded weekly. All experimental procedures were approved by the Marshall University Laboratory Animal Care and Use Committee and were under the Guide for the Care and Use of Laboratory Animals issued by the National Institutes of Health.

### 4.2. Treadmill Exercise Protocol

The hypertensive transgenic mice (R+ mice, male and female; 7–8 weeks old) were randomly assigned to a sedentary or exercise group. The exercise protocol was based on our previous study [20]. All mice in the exercise group were acclimatized to running on a motorized rodent treadmill with an electric grid at the rear of the treadmill (Columbus Instruments, Columbus, OH, USA) for 6 days before the 8-week exercise intervention was conducted. During acclimation, the treadmill duration was 5 m/min for 30 min on day 1. The speed was then increased by 1 m/min, and the time was increased by 10 min daily until the exercise training reached the target speed of 10 m/min and time of 60 min. The treadmill speed was set at 10 m/min for exercise intervention, and daily 60-min treadmill exercise was repeated for 8 weeks, 5 days per week.

### 4.3. EPC Culture and EPC-EX Generation

The EPCs were cultured from the bone marrow of sedentary and exercised hypertensive R+ mice (15–16 weeks old) following the published protocol [48]. The cell culture medium was replaced every two days. After 5 days of culture, the EPC medium was changed to serum-free EBM-2 medium supplemented with growth factors (Lonza, Morristown, NJ, USA), for 2 days to stimulate EX release. Then EXs were collected from the culture medium using the ultracentrifuge method we previously reported with slight modification [49]. In brief, the culture medium was collected and centrifuged at 300× *g* for 15 min, and the supernatant was centrifuged at 2000× *g* for 20 min to remove cell debris. Then the resulting supernatant samples were centrifuged at 20,000× *g* for 70 min to get rid of microvesicles, followed by an ultracentrifugation at 170,000× *g* for 90 min to pellet EXs.

The EXs collected from the EPCs of sedentary hypertensive R+ mice or exercised R+ mice were denoted as R-EPC-EX, or R-EPC-EX^ET^, respectively. All collected EXs were characterized by nanoparticle tracking analysis (NTA) and Western blot with exosome-specific markers, Tsg101 and CD63, as well as EPC-specific markers CD34 and VEGFR2, as reported [22,49].

### 4.4. Uptake Efficiency Analysis of EPC-EXs with N2a Cells in a Co-Culture System

The mouse neural N2a cells were purchased from ATCC and cultured with Eagle’s minimum essential medium (EMEM) supplemented with 10% FBS and 1× strep–penicillin antibiotic solution according to the manufacturer’s instructions.

To assess the incorporation efficiency by N2a cells, EPC-EXs were labeled with a fluorescent dye PKH26 (Sigma-Aldrich, St. Louis, MO, USA). In brief, after ultracentrifugation isolation, the EX pellets were resuspended with 500 μL PBS containing 1 μL PKH26 and incubated for 5 min at room temperature. Then the reaction was stopped by adding 500 μL 1% BSA/PBS and left for 5 min at room temperature. EXs were then collected from the exosome suspension by ultracentrifugation, resuspended with the N2a cell culture medium, and used for co-culture experiments.

For the co-culture study, the N2a cells growing in a 24-well plate were co-cultured with R-EPC-EX or R-EPC-EX^ET^. The dose of EPC-EXs (1 × 10^9^ EXs/microliter) was chosen based on our previous report (Wang et al., 2020) [20]. The fluorescence signal in the cytoplasm of N2a cells was observed using an inverted fluorescence microscope at 8 h, 16 h, and 24 h of co-incubation. The images were all taken with the same gain/intensity with the same threshold. In each group, four random images in each well were taken. The experiment was repeated three times. The fluorescence signal was analyzed by Image J (NIH, Bethesda, MD, USA). Data were normalized to the fluorescence signal of the R-EPC-EX at 24 h co-culture.

### 4.5. MiR Profiling Analysis of EPC-EXs

The total RNAs of EPC-EXs isolated from sedentary and exercised hypertensive R+ mice were extracted using the Trizol agent. RNA was eluted in 10 μL of RNase-free water. The concentration of RNA was measured by a Nanodrop 2000 spectrophotometer (Thermo Fisher Scientific, Waltham, MA, USA). The miR profiling of EXs was performed using the mouse miRNome microRNA Profilers QuantiMirTM (384-well plate) from System Biosciences (SBI, Mountain View, CA, USA) according to the manufacturer’s instruction. In brief, the QuantiMir RT kit was used to prepare cDNA from the eluted RNA. A total of 800 ng total RNA was mixed with 5× PolyA buffer, 25 mM MnCl2, 5 mM ATP and PolyA polymerase and incubated for 30 min at 37 °C, then the oligo dT adaptor was added. The mixture was heated for 5 min at 60 °C and followed by cooling to room temperature for 2 min. After that, 5× RT buffer, dNTP mix, 0.1 M DTT, and reverse transcriptase was added to the mixture to synthesize cDNAs at 42 °C for 60 min incubation, then heat for 10 min at 95 °C. Then the cDNA templates were ready for qPCR. For RT-qPCR reaction, 2× SYBR Green qPCR mastermix buffer (Applied Biosystems, Bedford, MA, USA) was mixed with the universal reverse primer (10 μM; provided by the mouse miRNome microRNA Profilers QuantiMirTM kit) and the synthetized QuantiMir cDNA templates for one entire 384-well qPCR plate. The SYBR green-based qPCR was performed in the QuantStudio 7 Flex instrument: 50 °C for 2 min, 95 °C for 10 min, 95 °C for 15 s, 60 °C for 1 min (40 cycles), data read at 60 °C for 1 min step. A set of 709 primers in the miR assay were used. Three endogenous reference RNAs (RNU1A, U6 snRNA, RNU43) were used as normalization signals for the miRNome microRNA profiling analysis. The result of miRNome profiling of each pair was processed using the ΔΔCT analysis software for mouse miRNome Ver. 14 (System Biosciences, Palo Alto, CA, USA), and the fold change > 3 was shown.

### 4.6. QRT-PCR Analysis of miR-27a in EPC-EXs and N2a Cells

The levels of miR-27a in R-EPC-EXs, R-EPC-EX^ET^, and N2a after co-incubation was determined by qRT-PCR, as we have reported [50]. In brief, the total miRs were extracted using the Trizol reagent, and the RNA concentration was measured using NanoDrop 2000 (Thermo Fisher Scientific). Then the cDNA was synthesized using the PrimeScript RT reagent kit (Takara Bio Inc., Kusatsu, Japan) according to the manufacturer’s instructions. The qRT-PCR was performed using the SYBR Premix Ex Taq kit (Takara Bio Inc.) on StepOne Plus. The RT primer for miR-27a was: 5′-GTC GTA TCC AGT GCA GGG TCC GAG GTA TTC GCA CTG GATACG AC TGC TCA-3′. The forward primer was: 5′-CAC GAA AGG GCT TAG CTG CTT GT-3′, and the reverse primer was 5′-CCA GTG CAG GGT CCG AGG TA-3′. U6 was applied as a housekeeping gene. The data were analyzed by the 2^−ΔΔCT^ method.

### 4.7. Co-Incubation of EPC-EXs with N2a Cells Challenged by Ang II Plus Hypoxia

To mimic the injury of hypertension and hypoxia in neurons, we challenged N2a cells (purchased from ATCC) with angiotensin II (Ang II) and hypoxia [51]. In brief, before being co-cultured with EPC-EXs, the culture medium of N2a cells in a 96-well plate was replaced with 100 μL serum-free glucose-free culture medium supplemented with Ang II (1 μM), then the cells were cultured in a hypoxia chamber (1% O_2_, 5%CO_2_, and 94%N_2_; Biospherixhypoxia chamber, NY) for 6 h, followed by 24 h reoxygenation in a standard CO_2_ incubator. During the reoxygenation period, the cells were assigned into different treatment groups: vehicle (no EX added), R-EPC-EX, or R-EPC-EX^ET^. To eliminate the potential effects of miR-27a, N2a cells were transfected with miR-27a inhibitors (0.1 nM; Dharmacon, Lafayette, CO, USA) with DharmaFECT transfection reagent (Dharmacon, Lafayette, CO, USA) [19] in the presence of R-EPC-EX^ET^. The cells cultured with EMEM supplemented with 10% FBS and 1× antibiotic solution were used as controls. The concentration of EXs (1 × 10^9^ EXs/microliter) was chosen based on a previous study [20]. After 24 h co-incubation, the cells were used for different assays described below.

### 4.8. Cell Viability Analysis of N2a Cells

The cell viability of N2a was assessed by MTT-(3-[4, 5-dimethylthiazyol-2yl]-2, 5-diphenyltetrazolium bromide, Sigma) assay [52]. After 24 h co-culture, the cell culture was replaced with 100 μL fresh culture medium, and 10 μL of the 12 mM MTT solution was added to each well. The cells were incubated for 4 h at 37 °C. Then, all but 25 μL of the medium was removed from each well, add 50 μL DMSO was added to each well and mixed by pipetting, and the plates were incubated for another 10 min at 37 °C. The absorbance of cells in each group was read on a microplate reader (BioTek, El Segundo, CA, USA) at 540 nm. The cells cultured with EMEM supplemented with 10% FBS and 1× antibiotic solution in a standard CO_2_ incubator were used as controls. Each group was duplicated, and the experiments were repeated three times. The percent cell viability was normalized to the control group (without Ang II and hypoxia challenge).

### 4.9. Dihydroethidium (DHE) Assay for Oxidative Stress Analysis of N2a Cells

Intracellular ROS generation is assessed to determine cell oxidative stress. For DHE staining, after 24 h co-culture with the various types of EPC-EXs, 10 μM DHE solution was added to the culture medium of each well of the N2a cells and the plates were incubated at 37 °C for 70 min. Then the medium was removed, and the cells were rinsed with PBS twice. The fluorescence signal of the cells was observed under a fluorescence microscope. The fluorescence intensity of four randomly chosen microscopic areas in each well was analyzed using Image J (NIH, Bethesda, MD, USA). The cells cultured with EMEM supplemented with 10% FBS and 1× antibiotic solution in a standard CO_2_ incubator were used as controls. The experiment was repeated three times. The images were all taken with the same gain/intensity with the same threshold. The relative fluorescence intensity was normalized to that of the control group (without Ang II and hypoxia challenge).

### 4.10. JC-1 Staining of Mitochondrial Membrane Potential (MMP) Analysis in N2a Cells

The MMP of N2a cells treated with various EXs was measured using lipophilic cationic dye JC-1 (1:1000, Invitrogen, Carlsbad, CA, USA). After 24 h of treatment, N2a cells were washed with PBS and incubated with 500 μL of medium containing JC-1 staining probe (0.5 μL; 5 μg/μL stock concentration) for 30 min at 37 °C. Then, the cells were washed with PBS and observed under an inverted fluorescence microscope (Nikon, Melville, NY, USA). N2a cells cultured with EMEM supplemented with 10% FBS and 1× antibiotic solution in a standard CO_2_ incubator were used as controls. The level of cellular fluorescence intensity of four random microscopic areas in each well was analyzed using Image J 1.53t (NIH, Bethesda, MD, USA). The relative MMP was calculated as the ratio of J-aggregate to monomer (590/520 nm). The data are expressed as a fold of the control cells. The images were all taken with the same gain/intensity with the same threshold.

### 4.11. Western Blot Analysis

After 24 h of treatment, the proteins of N2a cells in different groups were extracted with cell lysis buffer (Fisher Scientific, Waltham, MA, USA) supplemented with a complete mini protease inhibitor tablet (Roche, Basel, Switzerland). Then, the protein lysates were electrophoresed through SDS-PAGE gel and transferred onto PVDF membranes. The membranes were blocked with 5% non-fat milk for 1 h at room temperature and incubated with primary antibody against cytochrome c (1:400; Abcam, Boston, MA, USA), Nox4 (1:1000; Abcam, Boston, MA, USA), and β-actin (1:4000; Sigma, St. Louis, MO, USA) at 4 °C overnight. The next day, the membranes were washed and incubated with horse radish peroxidase-conjugated anti-rabbit or anti-mouse IgG (1:40,000; Jackson Immuno Research Lab, West Grove, PA, USA) for 1 h at room temperature. Blots were developed with enhanced chemiluminescence developing solutions, and images were quantified using ImageJ 1.53t software.

### 4.12. Statistical Analysis

All experiments were repeated three times. Data are expressed as the mean ± SD. Two-group comparison was analyzed by Student’s *t*-test. Multiple comparisons were analyzed by one- or two-way ANOVA followed by the Tukey post-hoc test. GraphPad Prism 9 was used for analyzing the data. For all measurements, *p* < 0.05 was considered statistically significant.

## 5. Conclusions

In conclusion, our data have demonstrated that exercise intervention can modulate the functions of EXs released by bone marrow-derived EPCs in hypertensive conditions. These exercise-intervened EPC-EXs can protect the neuronal mitochondria against hypertensive and hypoxia injury via their carried miR-27a. These findings will help to understand the mechanism of exercise intervention in brain ischemic injury and facilitate the establishment of novel therapeutic strategies.

## Figures and Tables

**Figure 1 ijms-25-01148-f001:**
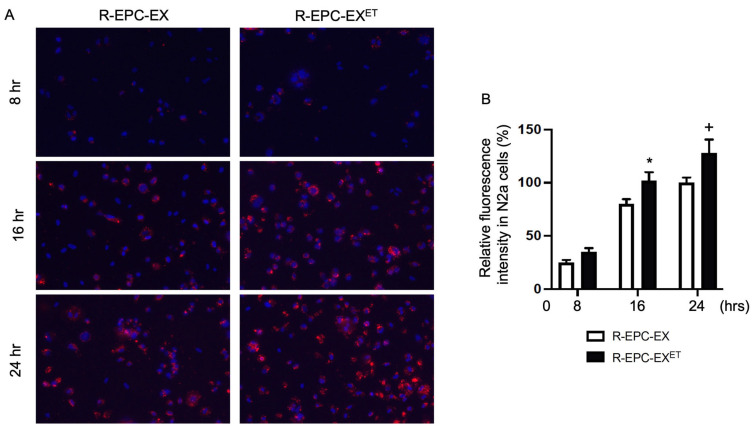
Exercise intervention improved the incorporation efficiency of EPC-EXs isolated from hypertensive transgenic mice by N2a cells. (**A**) representative images showing the incorporation of EX (labeled with PKH26, a red fluorescence) uptake by N2a cells at 8 h, 16 h, and 24 h incubation. Magnification: 200×. (**B**) summarized data showing the EPC-EX incorporation by N2a cells. Blue: DAPI. Red: PKH26 labeled EXs. * *p* < 0.05, vs. R-EPC-EX in 16 h; ^+^
*p* < 0.05, vs. R-EPC-EX in 24 h. Data expressed as mean ± SD, *n* = 3/group.

**Figure 2 ijms-25-01148-f002:**
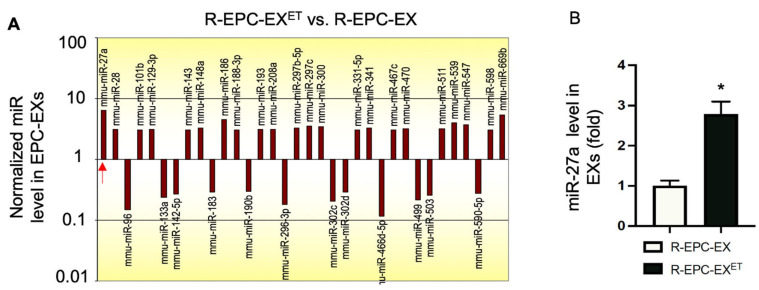
Profiling analysis of exosomal miRs and qRT-PCR of miR-27a in EPC-EXs of exercised and sedentary hypertensive mice. (**A**) representative miR profiling graph showing the changes of miRs in EPC-EXs. MiRs with down- or up-regulated >3-fold are listed. (**B**) qRT-PCR verified the miR-27a level in EPC-EXs. * *p* < 0.05, vs. R-EPC-EX. Data are expressed as mean ± SD, *n* = 3/group.

**Figure 3 ijms-25-01148-f003:**
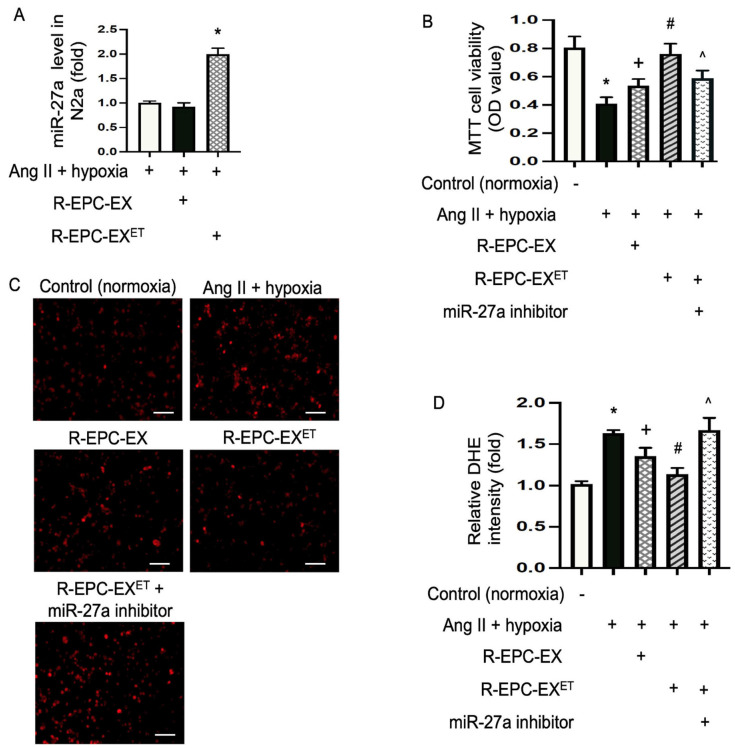
Exercise-intervened EPC-EXs raised miR-27a level, improved the cell survival ability, and reduced ROS overproduction in N2a cells challenged by Ang II plus hypoxia. (**A**) miR-27a level in N2a cells co-cultured with or without R-EPC-EXs. (**B**) summarized data showing the N2a cell viability assessed by MTT assay. (**C**) representative images of N2a cells of DHE staining (red fluorescence). Scale bar: 100 μm. (**D**) summarized data of DHE staining assay for ROS production in N2a cells. * *p* < 0.05, vs. con (no Ang II + hypoxia); ^+^
*p* < 0.05, vs. Ang II + hypoxia only; ^#^
*p* < 0.05, vs. R-EPC-EX; ^ *p* < 0.05, vs. R-EPC-EX^ET^. Data expressed as mean ± SD, *n* = 3/group.

**Figure 4 ijms-25-01148-f004:**
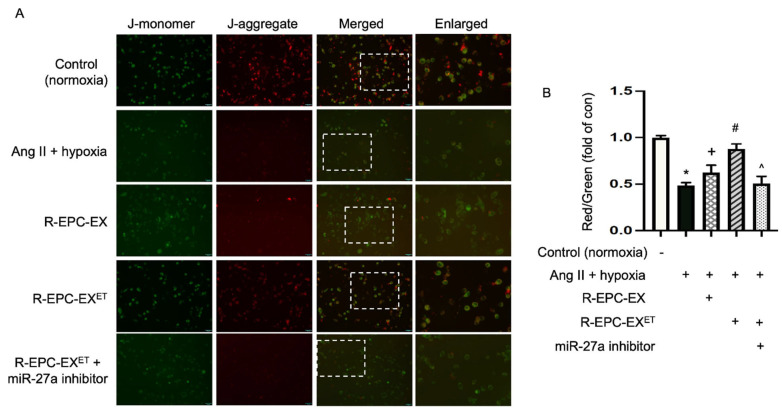
Exercise-intervened EPC-EXs regulated the MMP in N2a cells challenged by the Ang II plus hypoxia. (**A**) representative images show the JC-1 staining in N2a cells. Enlarged images are from the dashed box of the merged images. Scale bar: 50 μm. (**B**) summarized data of JC-1 stain. * *p* < 0.05, vs. con (no Ang II + hypoxia); ^+^
*p* < 0.05, vs. Ang II + hypoxia only; ^#^
*p* < 0.05, vs. R-EPC-EX; ^ *p* < 0.05, vs. R-EPC-EX^ET^. Data expressed as mean ± SD, *n* = 3/group.

**Figure 5 ijms-25-01148-f005:**
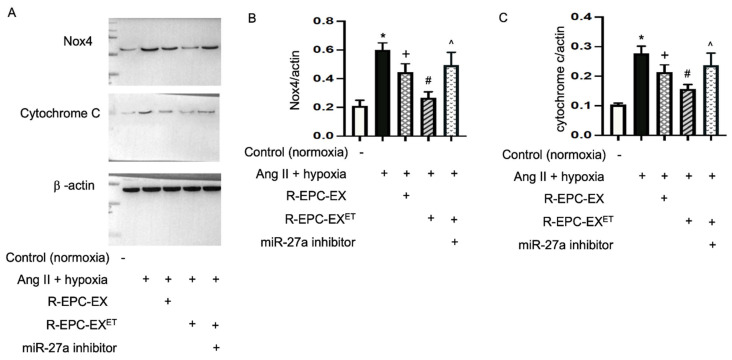
Exercise-intervened EPC-EXs regulated the expressions of Nox4 and cytochrome c in Ang II plus hypoxia-challenged N2a cells. (**A**–**C**) representative bands, and summarized data of cytochrome c and Nox4 levels in N2a cells. * *p* < 0.05, vs. con (no Ang II + hypoxia); ^+^
*p* < 0.05, vs. Ang II + hypoxia only; ^#^
*p* < 0.05, vs. R-EPC-EX; ^ *p* < 0.05, vs. R-EPC-EX^ET^. Data expressed as mean ± SD, *n* = 3/group.

## Data Availability

The datasets generated during the current study are available from the corresponding author upon reasonable request.

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
