# Peer review of "Exercise-Intervened Endothelial Progenitor Cell Exosomes Protect N2a Cells by Improving Mitochondrial Function"

_ijms, 2024, doi:10.3390/ijms25021148_

Round 1

Reviewer 1 Report

Comments and Suggestions for Authors

I liked the work, but I pointed out some things that need to be improved.

I found the arrangement of the results a little confusing. It can be improved in general.

The literature needs to be updated. Do you have any approval from the ethics committee for the use of animals? Briefly describe the isolation of exosomes. It is necessary to set the magnification used in microscopy. what is the RNA concentration nanodrop, and What concentration did they use to synthesize the cDNA? It needs to be better described, with times and temperatures, how many cycles... 

Standardize and use symbols for microliter and microgram.

Suggestions for results: do not include the methodology or a discussion part. Enter the result.

I would suggest a suggestion for discussion: start the paragraph with your result, then a discussion of the literature, and finally a sentence with a conclusion.

Author Response

Q1. I found the arrangement of the results a little confusing. It can be improved in general.

Response: We are sorry for making the reviewer feel the confusion. We have revised some of the results to make clear statements.

Q2. The literature needs to be updated. Do you have any approval from the ethics committee for the use of animals? Briefly describe the isolation of exosomes. It is necessary to set the magnification used in microscopy. what is the RNA concentration nanodrop, and What concentration did they use to synthesize the cDNA? It needs to be better described, with times and temperatures, how many cycles... 

Response: Yes, we did have approval for the animal usage. It is listed in a separate section titled “Ethical approval” (pages 556-559). The detailed protocol of exosome isolation has been added in the method section (lines 153-156). The magnification of the microscopy is 200x-400x. We provided the scale bar for each set of images. The RNA concentration was measured by Nanodrop 2000. The RT was conducted using the QuantiMir RT reaction kit following the manufacturer’s instructions. The QuantiMir RT reaction kit is included in the mouse miRNome microRNA Profilers QuantiMirTM kit. The detailed method has been added in the revision (lines 190-200).

Q3. Standardize and use symbols for microliter and microgram.

Response: Yes, the symbols for microliter and microgram are corrected.

Reviewer 2 Report

Comments and Suggestions for Authors

This article is interesting aiming to investigate whether exercise training over a period of 8 weeks can modifies the miR carried by exosomes originating from EPC. This article presents evidence that exosome derived from EPC see significant changes in their miRNA content with 30 miRs showing differential expression after training (8 weeks). The author identifies miR-27 as a key exercise-driven miR from EPC origins that could potentially limit negative impact (i.e. oxidative stress, mitochondrial defect and survival) of a dual treatment with angiontensin 2 and hypoxia (aiming here to mimic hypertension and ischemia simultaneously).

The article is valuable and interesting for the audience. Yet there are some limitations that might need to be address or requires more explanation.

While the authors have previously published with this model of hypertensive R+ mice. It seems still important to have some information regarding the level of hypertension observed in these mice and whether 8 weeks of training resulted in any changes in blood pressure. This information could be included in the text at the beginning of the results section or in figure 1.

It is unclear why the authors focus on miR-27 specifically and not other miRs that are highly differently abundant when comparing hypertensive alone versus exercised. For example, miR96 and miR-466d-5p are more downregulated than miR27 is upregulated. It cannot be excluded that miR96 and miR-466-5p could be support detrimental mechanisms through which hypertension challenge neuroblast health.

While the idea to treat the neuroblasts with angiotensin 2 and hypoxia is relevant. Background is missing for the reading to fully grasp key information regarding how this mimics a stroke or hypertension. Is it well described that neurons are exposed to more Ang-II and hypoxia simultaneously? While hypertension is often caused by dysregulation of the renin-angiotensin aldosterone system, the authors do not provide this information in the introduction to help the reader better understand the value of their approach. It will highly strengthen the article if an additional control was added in the experiment presented in figures 3, 4 and 5.

For more specific information please see the point below

Line 19 _ The authors specify that miR-27 expression is restored with the exercise intervention. But it was not noted specify before how hypertension or AngII+hypoxia altered this specific miR27. Editing might be needed to support a better flow of idea in the abstract.

Introduction:

Line 54 _ “Together these findings indicate the involvement of EPCs in hypertension-related 54 cardiovascular disease.” This statement is an extrapolation. Authors need to be more cautious. The references cited have investigated the impact of exercise on EPC mobilisation, VEG-production or angiogenesis not on blood pressure. Authors must revise their statement to ensure more accuracy with the references used.

Line 115: The neural cells used are the N2a which are neuroblast. Would it be more appropriate to refer to these cells are neuroblasts? It also appears important to specific that they are a mouse origin. As the author used mouse EPC as donor of exosome it is key that they also use mouse neuroblast as recipients of these mouse exosomes.

            The title implies that studies were done on neurons. Yet the methods do not indicate the N2A neuroblasts were differentiated in neurons before measurements were done. If the authors did not perform differentiation of N2a cells this would then mean that the conclusion of the paper are only valid from neuroblast cells. It will be crucial to clarify this point. Where the experiments ran on neuroblast or differentiated N2a = neuron-like cells?

If authors have differentiated the N2a cells how have they verify that these cells have acquired the expected neuronal phenotype in their culture?

Line 144: The authors indicate that they used 3 endogenous references. It will be valuable to specific how this was used for the normalization of the qPCR. Did the authors used a geometric mean to implement in the ddCt calculation? Methods need to be revised to provide more information.

Line 237: I understand that the authors treated with the same amount of exsomes. Yet, have the authors verify that the incorporation of PKH26 is similar between the 2 groups of exosomes when mixing the exosomes with the fluorescent label? Just to ensure that no biais could be due to more label incorporation in the exosome from exercise.

Line 255: Is or are references needed here to support the statement.

Line 262-263: “The results showed that the miR-27a levels were significantly 262 decreased in R-EPC-EXs, which was restored by exercise intervention (Fig 2B).” Here the authors have no mice that are normotensive. So, the wording is weird. It will be better to just state that miR-27a is greater after exercise intervention.

Figures 3, 4 and 5: this experiment is missing a control which is mR-27 inhibitor alone at least with the hypoxia+Ang II treatment. Because it cannot be excluded that miR27 is required for basal function and survival in the absence of any stressors.

Minor point: Is the use of "Exercise-intervened" best and appealing enouh for readers.

Author Response

Q1. While the authors have previously published with this model of hypertensive R+ mice. It seems still important to have some information regarding the level of hypertension observed in these mice and whether 8 weeks of training resulted in any changes in blood pressure. This information could be included in the text.

Response: Thank the reviewer for this point. In our exercise regimen, we have observed a decreased trend but no significant change in blood pressure after exercise intervention. We have added this information in the revision (lines 443-448).

Q2. It is unclear why the authors focus on miR-27 specifically and not other miRs that are highly differently abundant when comparing hypertensive alone versus exercised. For example, miR96 and miR-466d-5p are more downregulated than miR27 is upregulated. It cannot be excluded that miR96 and miR-466-5p could be support detrimental mechanisms through which hypertension challenge neuroblast health.

Response: We appreciate the reviewer for this point. We chose miR-27a because it is one of the potential neurovascular protective miRs. Previous studies have demonstrated that miR-27a can alleviate brain injury in hemorrhagic stroke and alleviate hypoxia-induced neuron apoptosis. It is also involved in mediating the protective effect of exercise on the aorta. All these findings indicate the implication of miR-27a in ischemic injury and participation in exercise intervention. We also agree with the reviewer that we are not able to exclude the potential effects of other miRs such as miR-96 or miR-466d on the neuroblast N2a cells challenged by hypertension. We only conducted miR profiling analysis in the EXs collected from the culture medium of EPCs and have not determined the levels of miR-96 or miR-466d on N2a cells challenged by Ang II. We added a discussion in the revision (lines 506-511).

Q3. While the idea to treat the neuroblasts with angiotensin 2 and hypoxia is relevant. Background is missing for the reading to fully grasp key information regarding how this mimics a stroke or hypertension. Is it well described that neurons are exposed to more Ang-II and hypoxia simultaneously? While hypertension is often caused by dysregulation of the renin-angiotensin aldosterone system, the authors do not provide this information in the introduction to help the reader better understand the value of their approach. It will highly strengthen the article if an additional control was added in the experiment presented in figures 3, 4 and 5.

Response: We revised the background section by taking the reviewer’s suggestion (lines 109-113 and 116-119). Additional control groups as suggested by the reviewer here and in Question 11 will be studied in future studies. We have included it in the revision (lines 504-506).

Q4. The authors specify that miR-27 expression is restored with the exercise intervention. But it was not noted specify before how hypertension or AngII+hypoxia altered this specific miR27. Editing might be needed to support a better flow of idea in the abstract.

Response: We appreciate the reviewer for this point. We measured the expression of miR-27a in EPC-EX from the wild-type mice and found that its level was downregulated in R-EPC-EXs. We added the data in the revision (lines 328-330).

Q5. “Together these findings indicate the involvement of EPCs in hypertension-related 54 cardiovascular disease.” This statement is an extrapolation. Authors need to be more cautious. The references cited have investigated the impact of exercise on EPC mobilisation, VEG-production or angiogenesis not on blood pressure. Authors must revise their statement to ensure more accuracy with the references used.

Response: By taking the reviewer’s suggestion, we have revised the statement in the introduction section (lines 68-77 and 84-85).

Q6. The neural cells used are the N2a which are neuroblast. Would it be more appropriate to refer to these cells are neuroblasts? It also appears important to specific that they are a mouse origin. The title implies that studies were done on neurons. Yet the methods do not indicate the N2A neuroblasts were differentiated in neurons before measurements were done. If the authors did not perform differentiation of N2a cells this would then mean that the conclusion of the paper are only valid from neuroblast cells. It will be crucial to clarify this point. Where the experiments ran on neuroblast or differentiated N2a = neuron-like cells? If authors have differentiated the N2a cells how have they verify that these cells have acquired the expected neuronal phenotype in their culture?

Response: Sorry for the missing information on the N2a cells. We have added the relative information (mouse origin) in the method section. We did not differentiate the N2a cells into neuron-like cells. By taking the reviewer’s suggestion, we revised the paper title to “Exercise-intervened endothelial progenitor cell exosomes protect N2a cells by improving mitochondrial function”.

Q7. The authors indicate that they used 3 endogenous references. It will be valuable to specific how this was used for the normalization of the qPCR. Did the authors used a geometric mean to implement in the ddCt calculation? Methods need to be revised to provide more information.

Response: Sorry for the confusion. The 3 endogenous references (RNU1A, U6 snRNA, and RNU43) were used for mouse miRNome profiling analysis as provided by the profiling kit. The data analysis was conducted using the ΔΔCT analysis software for mouse miRNome Ver.14 provided by the manufacturer (System Biosciences). We have added the information in the method section (lines 203-206).

Q8. I understand that the authors treated with the same amount of exsomes. Yet, have the authors verify that the incorporation of PKH26 is similar between the 2 groups of exosomes when mixing the exosomes with the fluorescent label? Just to ensure that no biais could be due to more label incorporation in the exosome from exercise.

Response: We appreciate the reviewer for this great point. We re-measured the concentration of exosomes after PKH26 labeling using the nanoparticle tracking analysis before co-culture experiments.

Q9. Is or are references needed here (line 255) to support the statement.

Response: We have added references to support the statement lines (322-323).

Q10. Line 262-263: “The results showed that the miR-27a levels were significantly 262 decreased in R-EPC-EXs, which was restored by exercise intervention (Fig 2B).” Here the authors have no mice that are normotensive. So, the wording is weird. It will be better to just state that miR-27a is greater after exercise intervention.

Response: The expression of miR-27a in EPC-EXs from hypertensive mice was downregulated as compared to that from wild-type mice. We revised the statement (lines 328-332). Please also refer to Question 4.

Q11. Figures 3, 4 and 5: this experiment is missing a control which is mR-27 inhibitor alone at least with the hypoxia+Ang II treatment. Because it cannot be excluded that miR27 is required for basal function and survival in the absence of any stressors.

Response: We appreciate the reviewer for this point. Previous studies show that miR-27a can alleviate hypoxia-induced neuronal apoptosis in mouse brains. We will determine the effects of miR-27a in N2a cells in unstressed conditions in future studies (lines 504-506).

Q12. Minor point: Is the use of "Exercise-intervened" best and appealing enough for readers.

Response: The major purpose of this study is to compare the function difference of EPC-EXs from exercised and non-exercised hypertensive mice. We think the term “exercise-intervened” is appropriate to describe the status of EPC-EXs from the exercised mice.

Round 2

Reviewer 2 Report

Comments and Suggestions for Authors

I would like to thank the authors for taking in consideration my comments and addressing them efficiently.

I do not have any further comments.